# Promising Natural Compounds against Flavivirus Proteases: Citrus Flavonoids Hesperetin and Hesperidin

**DOI:** 10.3390/plants10102183

**Published:** 2021-10-14

**Authors:** Raphael J. Eberle, Danilo S. Olivier, Marcos S. Amaral, Dieter Willbold, Raghuvir K. Arni, Monika A. Coronado

**Affiliations:** 1Institute of Biological Information Processing (IBI-7: Structural Biochemistry), Forschungszentrum Jülich, 52428 Jülich, Germany; D.Willbold@fz-juelich.de; 2Institut für Physikalische Biologie, Heinrich-Heine-Universität Düsseldorf, Universitätsstraße, 40225 Düsseldorf, Germany; 3Integrated Sciences Center, Federal University of Tocantins, Araguaína 77824-838, Brazil; doliviercg@gmail.com; 4Institute of Physics, Federal University of Mato Grosso do Sul, Campo Grande 79070-900, Brazil; marcossamaral@gmail.com; 5JuStruct: Jülich Centre for Structural Biology, Forchungszentrum Jülich, 52428 Jülich, Germany; 6Multiuser Center for Biomolecular Innovation, Departament of Physics, Universidade Estadual Paulista (UNESP), São Jose do Rio Preto 15054-000, Brazil; raghuvir.arni@unesp.br

**Keywords:** hesperetin, hesperidin, flavivirus, NS2B/NS3^pro^, noncompetitive inhibitors

## Abstract

Ubiquitous in citrus plants, Hesperidin and Hesperetin flavanones possess several biological functions, including antiviral activity. Arbovirus infections pose an ever-increasing threat to global healthcare systems. Among the severe arboviral infections currently known are those caused by members of the Flavivirus genus, for example, Dengue Virus—DENV, Yellow Fever Virus—YFV, and West Nile Virus—WNV. In this study, we characterize the inhibitory effect of Hesperidin and Hesperetin against DENV2, YFV, and WNV NS2B/NS3 proteases. We report the noncompetitive inhibition of the NS2B/NS3^pro^ by the two bioflavonoids with half maximal inhibitory concentration (IC_50_) values <5 µM for HST and <70 µM for HSD. The determined dissociation constants (K_D_) of both flavonoids is significantly below the threshold value of 30 µM. Our findings demonstrate that a new generation of anti-flavivirus drugs could be developed based on selective optimization of both molecules.

## 1. Introduction

Polyphenols are plant secondary metabolites that are encountered in fruits, vegetables, and seeds that are constituents of normal diet [1]. They occur widely in nature, and to date, more than 9000 flavonoids have been identified [2]; these include, but are not limited to, phenolic acids, coumarins, flavonoids, stilbenes, and lignans [3]. They are characterized by the presence of at least two phenolic groups associated in a complex structure. The wide spectrum of physiological functions exhibited by flavonoids includes antiviral/bacterial, anti-inflammatory, cardioprotective, antidiabetic, anticancer, and antiaging activities [4,5,6,7].

Both bioflavonoids HSD and HST are encountered in citrus fruits. Hesperidin (HSD) (3′,5,7-trihydroxy-4′-methoxy-flavanone-7-rhamnoglucoside) is a flavanone glycoside (flavonoid subclass) that consist of an aglycone, HST, and a disaccharide rutinose (Figure 1). Hesperetin (HST) (3′,5,7-trihydroxy-4′-methoxyflavanone) is its aglycone form [8]. 

Both flavonoids show antiviral activity and have been reported to inhibit the intracellular replication of Chikungunya and Sindbis viruses [9,10]. Recently, we showed the potential for HST and HSD to be used against Zika and Chikungunya virus proteases (ZIKV and CHIKV) [11].

Factors including urbanization, climate and ecological changes, and unsustainable vector control, make the diseases caused by arthropode transmitted viruses (arboviruses) more prevalent; therefore, they are becoming another emerging threat to global health and welfare [12,13]. Members of the Flavivirus genus cause the most severe arbovirus infections. The genus includes major human pathogens such as Yellow Fever virus (YFV), Dengue virus (DENV), Japanese encephalitis virus (JEV), tick-borne encephalitis virus (TBEV), and West Nile virus (WNV) [14,15,16]. Flaviviruses are single-stranded, positive-sense RNA viruses, whose genome encodes a polyprotein that can be co- and post-translationally cleaved into three distinct structural proteins and seven nonstructural (NS) proteins (NS1, NS2A, NS2B, NS3, NS4A, NS4B, and NS5) [17]. Important for the proteolytically procession of the polyprotein is the viral protease NS3 [18], which requires the NS2B cofactor for optimal enzymatic activity [19]. The flavivirus replication route is dependent on the NS2B/NS3 protease (NS2B/NS3^pro^), which represents a potential drug target against flavivirus infections [20,21].

Based on our previous findings on the inhibitory effect HST on ZIKV NS2B/NS3^pro^, we hypothesized that the HST and HSD molecules should also exert a similar inhibition of the NS2B/NS3 proteases of DENV2, YFV, and WNV. This hypothesis led us to believe that these molecules could be used as lead molecules in the development of a broad-spectrum antiviral. A fluorogenic inhibition assay demonstrated that HST inhibited the proteases with half-maximal inhibitory concentration (IC_50_) values in the low µM range, unlike the inhibition presented by HSD, which presented a IC_50_ value ten times higher. The determined dissociation constants (K_D_) for HST and HSD were <30 µM and docking, followed by MD simulations of HST and HSD in a complex with the proteases, suggested a possible inhibition mode of these molecules.

## 2. Results and Discussion

### 2.1. Preparation of Flavivirus Proteases

The target proteins (DENV2, YFV, and WNV NS2B/NS3^pro)^ were expressed in *E. coli* Lemo (DE3) cells. The soluble proteases were purified in two steps as follows: Ni-NTA to separate the target protein from *E. coli* proteins followed by subsequent size exclusion chromatography. The purity of the proteases was checked by SDS–PAGE 15% (Appendix A).

### 2.2. Inhibition of DENV2, YFV, and WNV Proteases by HST and HSD

Flavonoids and their subgroups comprise structural diversity, which can be used to identify potential inhibitors that can serve as lead compounds for the development of efficacious and efficient alternatives to combat pathogens [22]. Based on this, we performed a primary inhibitory test using HST and HSD to check their effect against DENV2, YFV, and WNV NS2B/NS3^pro^ (Figure 2).

An amount of 10 µM of HST inhibited the DENV2 NS2B/NS3^pro^ activity by around 55%, the YFV NS2B/NS3^pro^ (~80%), and the WNV NS2B/NS3^pro^ (~70%). In contrast, 10 µM of HSD shows inhibition of the three tested proteases to be around 30%. 

For further information of the inhibitory effects of both flavonoids, new experiments were carried out. To evaluate the effect of HST against the described viral proteases, the concentration range of 0–140 µM (DENV2) and 0–100 mM (YFV, WNV) were tested (Figure 3).

HST inhibited 100% of the DENV2 protease activity at a concentration of 140 µM (Figure 2A), and the activity of the YFV and WNV NS2B/NS3^pro^ were inhibited completely at a concentration of 80 µM (Figure 2C,E). The calculated IC_50_ values for HST and the virus proteases NS2B/NS3^pro^ are < 5 µM, DENV2 (4.7 ± 0.8 µM), YFV (2.0 ± 0.5 µM), and WNV (4.3 ± 1.6 µM) (Table 1 and Appendix A).

On the contrary, HSD inhibited the DENV2 and YFV protease activity by 100% at a concentration of 140 µM (Figure 4A,C), and the WNV protease was inhibited completely at a final concentration of 120 µM (Figure 3E), showing a probable steric effect of the sugar group. The calculated IC_50_ values for HSD and the virus proteases NS2B/NS3^pro^ are between 50 and 70 µM: DENV2 (55.7 ± 2.5 µM), YFV (67.7 ± 7.5 µM), and WNV (50.7 ± 8.3 µM) (Table 1 and Appendix A). 

Further experiments identified HST and HSD as noncompetitive inhibitors (Table 1 and Figure 3 and Figure 4B,D,F), as was described previously for ZIKV NS2B/NS3^pro^ [11]. As noncompetitive inhibitors, HST and HSD show a type of allosteric inhibition. 

When a noncompetitive inhibitor is added, the maximum rate (V_max_) values for the flavivirus NS2B/NS3 proteases were changed, when in the presence of HST and HSD at various concentrations; whereas, the Michaelis–Menten constant (K_M_) remains unaltered, which is shown on the Lineweaver-Burk plot by a change in the slope and y-intercept (see Figure 3 and Figure 4). In contrast, a competitive inhibitor would not affect the V_max_ values but would increase the K_M_ values [23].

We have previously reported that the IC_50_ value of HST for ZIKV NS2B/NS3^pro^ [11]—another flavivirus protease—was 12.6 ± 1.3 µM, which is in the same concentration range as the determined IC_50_ values of HST for DENV2, YFV, and WNV NS2B/NS3^pro^ (Table 1). The inhibitory effect of HSD against ZIKV NS2B/NS3^pro^ was not further investigated.

The inhibitory effect of bioactive flavonoids against flavivirus NS2B/NS3 proteases have already been described; furthermore, the IC_50_ values of our results by both molecules were in the same µM range as demonstrated by other flavonoids [11,24,25,26].

The measurement of ligand efficiency (LE) is a widely used tool for selecting leads for further improvement and development in examining the relationships between inhibitory potential, ligand binding affinity, and molecular size [27,28]. HST possesses LE values > 0.30, which indicates it to be an efficient binder to the tested virus proteases [27], with the potential to be used as a lead compound for further improvement and development of a broad-spectrum antiviral to fight flavivirus infections targeting the NS2B/NS3 protease. On the contrary, the LE values for HSD are below the method limit, which may depend on the size of the ligand (this has already been described in the literature) [28]. Our findings demonstrated that HST was the compound that consistently showed significant activity against all three tested virus proteases. The increased values of IC_50_ for HSD demonstrated the direct effect of the sugar moiety presented in the molecule. LE values clearly demonstrated the efficacy of HST as a lead compound against the studied flaviviruses.

### 2.3. DENV2, YFV, and WNV NS2B/NS3^pro^–HST and HSD Interaction, as Determined by TFS

TFS (tryptophan fluorescence spectroscopy) was used to characterize the studied viral proteases binding to HST and HSD. The viral proteases contain five (DENV2), seven (YFV), and six (WNV) tryptophan residues in the NS2B cofactor domain and in the NS3 protease domain. The maximum of the tryptophan fluorescence emission peak of the NS2B/NS3^pro^ occurred in ranges 350–361 nm (DENV2), 336–358 (YFV), and 340–355 (WNV), respectively, after titration with HST (Appendix A). Titration of the increasing amount of HST resulted in a red edge excitation shift (REES) of 11, 22, and 15 nm, respectively, of the proteases maximum fluorescence peaks to a lower wavelength. Increasing interactions between tryptophan and the surrounding solvent is characteristic for REES [29]. On the contrary, HSD interaction induced no shift of the fluorescence maximum (Appendix A).

The results of the TFS experiments allowed us to determine the K_D_ values of the proteases–flavonoid interaction (Appendix A). The calculated K_D_ values are summarized in Table 2. A similar binding strength between YFV and WNV NS2B/NS3^pro^ and HST could be observed (12.3 ± 2.2 µM and 13.5 ± 2.0 µM, respectively). The difference to the K_D_ value for the DENV2 NS2B/NS3^pro^–HST interaction is marginal at 18.7 ± 2.7 µM. The K_D_ values are similar to the ZIKV NS2B/NS3^pro^–HST interaction described previously (17.8 ± 2.9) [11]. 

The K_D_ value for the HSD interaction with DENV2 NS2B/NS3^pro^ is similar to the HST K_D_ (Table 2). In the cases of YFV and WNV NS2B/NS3^pro^, the HSD values increase to about factor 1.4 and 2.4, compared with HST (Table 2).

The NS2B/NS3 proteases of DENV2, YFV, and WNV share high sequences and structural similarities, as well as demonstrating a similar binding specificity towards HST. The increased values of K_D_ for the binding of HSD is probably due to the difference in its ability to accommodate the sugar moiety.

### 2.4. In Silico Studies of the Proteases–Flavonoids Interaction 

Prior to performing molecular docking, we carried out 200 ns of MD simulation of the single experimental models of DENV2 NS2B/NS3^pro^ (PDB entry: 4M9T), YFVNS2B/NS3^pro^ (PDB entry: 6URV), and WNV NS2B/NS3^pro^ (PDB entry: 2FP7). The flexibility of the protease structures of the MD simulation system was monitored by calculating the RMSD, RMSF, RoG, and the surface area (Appendix A). During the simulation of the DENV2, YFV, and WNF NS3^pro^ structures, the Cα RMSD for NS3^pro^ was below 3.5 Å for most of the simulation time. This was also the case for YFV and WNV NS2B structures, unlike the DENV NS2B Cα RMSD, which was about 5 Å. Clearly, the structures undergo some structural changes over the simulation time; however, simulations demonstrated stable structures with 200 ns (Appendix A).

In previous work, a hydrophobic allosteric pocket of NS3^pro^, located near the NS2B cofactor interaction area had been mapped for ZIKV NS2B/NS3^pro^ [11]. The same area was assumed for DENV2, YFV, and WNV proteases. Molecular docking and MD simulations of HST and HSD with the virus proteases were performed, aiming to identify a possible allosteric binding area. Therefore, HST and HSD were docked nearby the NS2B cofactor interaction area and two sets of simulations (200 ns each) were performed independently. Based on binding energy and frames in the cluster, a representative structure was chosen for each protease–ligand complex (Table 3).

The ligand–protein complexes analyses—through structural fluctuation—showed stability over the 200 ns of both molecular dynamics simulations as shown for: HST (Appendix A) and HSD (Appendix A). The position of both flavonoids in the hydrophobic allosteric pocket of NS3^pro^ is demonstrated for all three proteases in Appendix A.

Based on energy interactions, further analysis of the MD simulations identified the DENV2, YFV, and WNV NS2B/NS3^pro^ amino acid residues involved in the binding with HST and HSD (Figure 5 and Figure 6). Outstanding in the interaction of the virus proteases with HST is the region surrounding Ile123 (DENV2 and WNV) and Ile126 (YFV). In contrast to DENV2 and YFV NS2B, in the WNV NS2B/NS3^pro^ the residues Leu31 and Phe37 of the NS2B cofactor are involved in the interaction with HST (Figure 5C).

The two independent MD simulation replicas demonstrate a very similar location for HST in DENV2 and YFV NS2B/NS3^pro^. It is important to mention that the molecule was in the same starting poses; however, they had different velocities that implied different energies. The interacting amino acid residues were all the same, showing only minor intensity changes in energy due to statistical fluctuations along the trajectory (Figure 5A,B, and Appendix A). Contrary to this, during the MD simulation replicas of HST with the WNV protease, some fluctuations could be observed over time (Figure 5C and Appendix A). 

Similarly to HST, the surrounding areas of Ile123 (DENV2 and WNV) and Ile126 (YFV) were involved in the interaction with HSD, and the Ile residue showed mostly the strongest binding energy (Figure 6). In the YFV NS2B, four amino acid residues could be identified that stabilized the interaction with HSD (Leu31, Ser32, Glu33, and Phe37) (Figure 5B and Appendix A). Moreover, this region is involved in the WNV NS2B–HST interaction (Figure 5C).

The results of the two MD replicas of the protease–HSD complexes demonstrated that the fluctuation of HSD was much more dynamic when compared with the results presented by HST. The fluctuation of the sugar moiety (rutinose) demonstrates that it is not involved in a stable interaction with the proteases, e.g., a hydrogen bond. However, the results of the two independent runs of MD simulations allow us to assume that the regions surrounding Ile123 (DENV2 and WNV) and Ile126 (YFV) forms a potential allosteric binding site for HST and minor HSD. In our previous results with the ZIKV NS2B/NS3^pro^–HST complex, the MD simulations demonstrated the residues Leu31, Phe37, and Ile24 interacting with HST [11], which is in agreement with the presented results.

## 3. Materials and Methods

### 3.1. Cloning of DENV2, YFV, and WNV Proteases

DENV2 and WNV NS2B/NS3^pro^ constructs were provided by Prof. Rolf Hilgenfeld, University Lübeck, Germany. The cDNA encoding DENV2 and WNV NS2B/NS3^pro^ (GenBank Protein Accession number AHZ13508.1 and AAA48498.2) were synthesized and implemented in the ampicillin resistant vector pET-15b (+) (Fisher Scientific Geneart, Regensburg, Germany).

YFV NS2B/NS3^pro^ cDNA (GenBank Protein Accession number AAY34247.1, isolate Angola/14FA/1971) was codon optimized, synthesized (BioCat GmbH, Heidelberg, Germany), and implemented in the kanamycin resistant vector pET-24a (+).

The construct contains the NS2B cofactor region (DENV2: residues 49–96; YFV: 48–96; WNV: 52–96) linked with NS3 protein (DENV2 residues 1–178; YFV: 1–191; WNV: 1–184). Both domains are connected by a G4SG4 linker. The constructs contain an N-terminal hexahistidine affinity tag and a TEV proteinase cleavage site (ENLYFQG).

### 3.2. Heterologous Gene Expression and Purification of the Virus Proteases

The virus NS2B/NS3 plasmids of DENV2 (pET-15b (+)), YFV (pET-24a (+)), and WNV (pET-15b (+)) were transformed into Lemo (DE3) *E. coli* competent cells (New England BioLabs, Ipswich, MA, USA), which were cultured overnight in LB medium at 37 °C in the presence of appropriate antibiotics according to the vector. Upon reaching an optical density of ~0.8 at 600 nm, protein expression was induced by adding Isopropylthiogalactoside. Cells were harvested, lysed, and centrifuged at 8000 rpm for 90 min. The supernatant was collected and applied to an affinity column using immobilized metal (Ni-NTA). The target protein was eluted from 100 mM to 500 mM imidazole buffer. Subsequently, the proteins were further purified with size-exclusion chromatography using a Superdex 75 10/30 column and showed ~95% purity on SDS–PAGE (for detailed information see Appendix A). 

### 3.3. Inhibition Assay of DENV2, YFV, and WNV NS2B/NS3^pro^

The DENV2, YFV, and WNV NS2B/NS3^pro^ activity assay was performed as described previously [30,31,32], using a fluorogenic substrate (Boc–Gly–Arg–Arg–AMC; BACHEM, Bubendorf, Switzerland). This procedure was used to investigate the inhibitory effect of HST (Merck, Darmstadt, Germany, purity > 95%) and HSD (Merck, Darmstadt, Germany, purity > 80%) against the virus proteases. The DENV2 (250 nM), YFV (6 nM), and WNV (50 nM) NS2B/NS3^pro^, separately, were incubated with 0–140 µM HST, 0–160 µM HSD (DENV2 NS2B/NS3^pro^) and 0–100 µM HST, 0–140 µM HSD (YFV and WNV NS2B/NS3^pro^) in assay buffer containing 20 mM Tris pH 8.5, 10% glycerol, and 0.01% Triton X-100 and incubated for 1 h at RT.

The inhibition assay was performed in Corning 96-well plates (Merck, Darmstadt, Germany). The measurement started by addition of the substrate with a final concentration of 50 µM. The fluorescence intensities were measured at 60 s intervals over 30 min at 37 °C using an Infinite 200 PRO plate reader (Tecan, Männedorf, Switzerland). The excitation and emission wavelengths were 380 nm and 465 nm, respectively. The IC_50_ values were calculated using GraphPad Prism5 software (San Diego, CA, USA). All measurements were performed in triplicate, and data are presented as mean ± SD.

### 3.4. Characterisation of the Type of Inhibition and LE of HST and HSD

The inhibition mode of HST and HSD for viral proteases was characterized using a modified form of the activity assay described above. DENV2, YFV, and WNV NS2B/NS3^pro^ at concentrations of 250 nM, 6 nM, and 50 nM, respectively, were incubated with different concentrations of HST and HSD, separately, for 1 h at RT. The reaction was initiated by the addition of the corresponding concentration series of the substrate and the fluorescence intensities were measured at 60 s intervals over 30 min at 37 °C using an Infinite 200 PRO plate reader (Tecan, Männedorf, Switzerland). A Lineweaver-Burk approach was used to analyse the data, comparing the reciprocal of velocity (1/V) vs. the reciprocal of the substrate concentration (1/[S]) [24,33]. All measurements were performed in triplicate, and data are presented as mean ± SD.

The ligand efficiencies for HST and HSD were calculated using Formula (1) as follows [27]:(1.4x*p*IC_50_)/N(1)

*p*IC_50_ was obtained utilizing an online tool [34], and N is the number of all atoms except hydrogen.

### 3.5. Intrinsic Tryptophan Fluorescence Spectroscopy (TFS)

The dissociation constants (Kd) for HST and HSD in complexes with DENV2, YFV, and WNV NS2B/NS3^pro^ were determined using intrinsic tryptophan fluorescence spectroscopy (TFS), as described previously [35]. TFS was measured with a QuantaMaster40 spectrofluorometer (PTI, Birmingham, AL, USA), using 1 cm path length quartz cuvettes (105.253-QS, Hellma, Mühlheim, Germany). To avoid excitation of tyrosin residues, an excitation wavelength at 295 nm was chosen. The emission spectrum was collected in the range of 300–500 nm with the increment 1 nm. Each data point on the emission spectrum is an average of 10 accumulations. The final viral proteases concentration was set to 10 μM in a buffer containing 25 mM Tris-HCL, pH 8.5, 150 mM NaCl, and 5% glycerol. The measuring volume was 50 µL.

During the interaction measurement, the protein solution within the cuvette was titrated stepwise with a ligand stock solution (0.5 mM ligand + 10 μM protein): DENV NS2B/NS3^pro^–HST/HSD (0–70 µM); YFV NS2B/NS3^pro^–HST/HSD (0–100 µM); WNV NS2B/NS3^pro^–HST/HSD (0–100 µM).

The quenching of the protease fluorescence, ΔF (F^max^ − F), at 337 nm of each titration point was considered and a saturation binding curve was fitted using a nonlinear least squares fit procedure, [36], based on Equation (2) as follows [37]: Y = B_max_[Q]/Kd + [Q](2)
where [Q] is the ligand concentration in solution (quencher), Y is the specific binding derived by measuring fluorescence intensity, B_max_ is the maximum amount of the complex protease–ligand at saturation of the ligand, and Kd is the equilibrium dissociation constant. The percentage of bound protease (i.e., y) that is derived from the fluorescence intensity maximum is plotted against the ligand concentration.

A modified Hill equation was used to determine the Kd value, following relation (3) as follows [38,39]:Log (F − F^min^)/(F) = m log Kd + n log [Q](3)
where F^min^ is the minimal fluorescence intensity in the presence of the ligand and Kd is the equilibrium constant for the protein–ligand complex. The “binding constant” K is defined as the reciprocal of Kd, m is the Hill equation’s coefficient, and n is the number of occupied binding sites.

### 3.6. Statistical Analysis

Statistical analyses were performed with GraphPad Prism software version 8 (San Diego, CA, USA). The experimental measurements consisted of three independent replications. Data are presented as mean values ± standard deviation (SD). Statistical significance was analyzed using one-way ANOVA, followed by Tukey’s multiple comparison test. Significant differences were considered at *p* < 0.05 (*), *p* < 0.01 (**), and *p* < 0.001 (***).

### 3.7. Molecular Dynamics and Computational Analysis 

#### 3.7.1. Ligand Parameterization

The ligand structures for HST and HSD were retrieved from the Zinc database [40]. Ligands were parameterized for MD simulation using Gaussian16 [41] at the B3LYP/6–31G* level of theory. The geometry was optimized, and the electrostatic potentials were calculated. Antechamber [42] was used to determine the restrained electrostatic potential (RESP) charges, and the general amber force field (GAFF) [43] was used for missing parameters. 

#### 3.7.2. System Preparation

The viral protease structures were retrieved from the PDB database with the following PDB codes: DENV—4M9T, YFV—6URV, and WNV—2FP7. All proteases, single or in complexes (protease–ligand), were placed in an octahedral box of TIP3P water extended to at least 10 Å of any solute atom. The systems were neutralized with Na^+^ or Cl^−^ ions. The proteins were submitted to H++ web server [44] to adjust the lateral sidechain of the amino acids to simulate at pH 7.4. 

#### 3.7.3. Simulation Setup

Amber 18 [45] was used to carry out all MD simulations. Protein interactions were described using the all-atom force field FF19SB [46], while the GAFF and RESP charges were used to describe the ligand molecules. To remove bad contacts from the initial structures, each system was energy minimized in two steps. First, the energy minimization of the restricted protein or complex was performed with 5000 steepest descending steps, followed by 5000 conjugated gradient steps, with a force constant of 10.0 kcal/mol-Å^2^. A second round of unconstrained energy minimization was performed during 10,000 steps. After minimization, the system was heated from 0 to 298 K for 500 ps under constant atom number, volume, and temperature (NVT) ensemble, with proteins constrained with force constant of 10 kcal/mol-Å^2^. 

Equilibration processes were performed using constant atom number, pressure, and temperature (NPT) ensemble, divided into six steps with decreasing force constant constraint of the protein or complex atoms from 10 to 0 kcal/mol-Å^2^. Finally, a production run was performed for each system for 200 ns in an NVT ensemble without any restriction. All simulated systems were duplicated. The SHAKE restrictions were applied to all bonds involving hydrogen atoms to allow for a 2 fs dynamic time interval. The long-range electrostatic interactions were calculated using the Ewald particle mesh (PME) method using 8 Å cutoff [47]. Langevin coupling was used to control temperature (298 K) and pressure (1 atm). 

#### 3.7.4. Molecular Dynamics Analysis

MD results were analyzed using CPPTRAJ [48] tools of the AmberTools19 [49] package. Equilibration and convergence of the systems (single protein and complexes) were investigated with root-mean-square Deviation (RMSD). To quantify protein flexibility, the root-mean-square Fluctuation (RMSF) of the Cα atoms were calculated. Structural changes in the protein structure were evaluated by determination of the radius of gyration (RoG) and surface area.

Representative structures for each simulation were obtained using clustering analysis with a k-means method, ranging from 2 to 6 clusters, while the clustering quality was accessed using the Davies–Bouldin index (DBI) values. 

The molecular mechanics/generalized Born surface area (MM/GBSA) was calculated between protein–ligand complexes using the generalized Born (GB)-Neck2 [50] implicit solvent model (igb = 8), in the steady-state regime of the last 100 ns of the simulation time, stripping the solvent and ions.

## 4. Conclusions

Currently, no specific therapy can significantly inhibit infection caused by DENV, YFV, and WNV viruses. Treatment focuses on alleviating symptoms; however, they do not treat the infection itself. Exploring promising antiviral agents remains essential for the cure of flavivirus infection. In this study, we demonstrated that Hesperidin and its aglycone form, Hesperetin—bioactive citrus plant flavonoids—might provide benefits in the development of a new drug to fight DENV2, YFV, and WNV, via inhibition of the NS2B/NS3 protease. 

Our results can be considered as a preliminary investigation and specific modifications can increase the relevant pharmaceutical properties of the molecules; for example, specificity towards targets, absorption of molecules into the cell and passage through the cell wall, as well as the metabolisms of the molecules in the cell.

## Figures and Tables

**Figure 1 plants-10-02183-f001:**
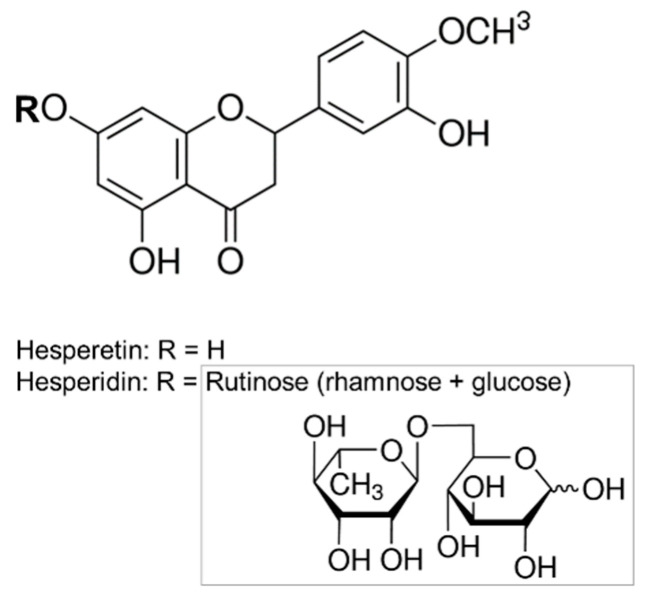
The chemical structures of Hesperetin and Hesperidin. The grey box shows the chemical structure of Rutinose.

**Figure 2 plants-10-02183-f002:**
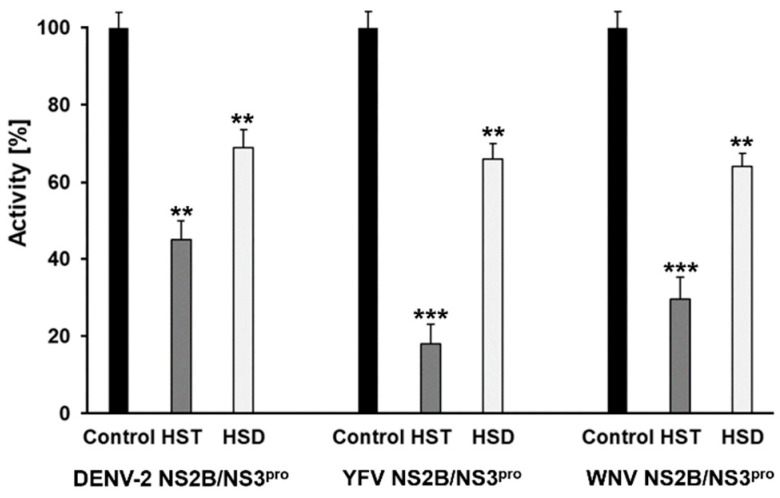
Primary inhibition test using HST and HSD as inhibitor molecules against flavivirus proteases. DENV2, YFV, and WNV NS2B/NS3^pro^ enzymatic activity [%]. Final concentration of the compound was set to 10 µM. HST inhibited the proteases activity between 55% and 80%. Contrary to this, HSD inhibited the proteases activity by about 30%. The data shown are the mean ± SD from 3 independent measurements (*n* = 3). Asterisks mean that the data differs from the control (0 µM inhibitor) significantly at *p* < 0.01 (**) and *p* < 0.001 (***), level according to ANOVA and Tukey’s test.

**Figure 3 plants-10-02183-f003:**
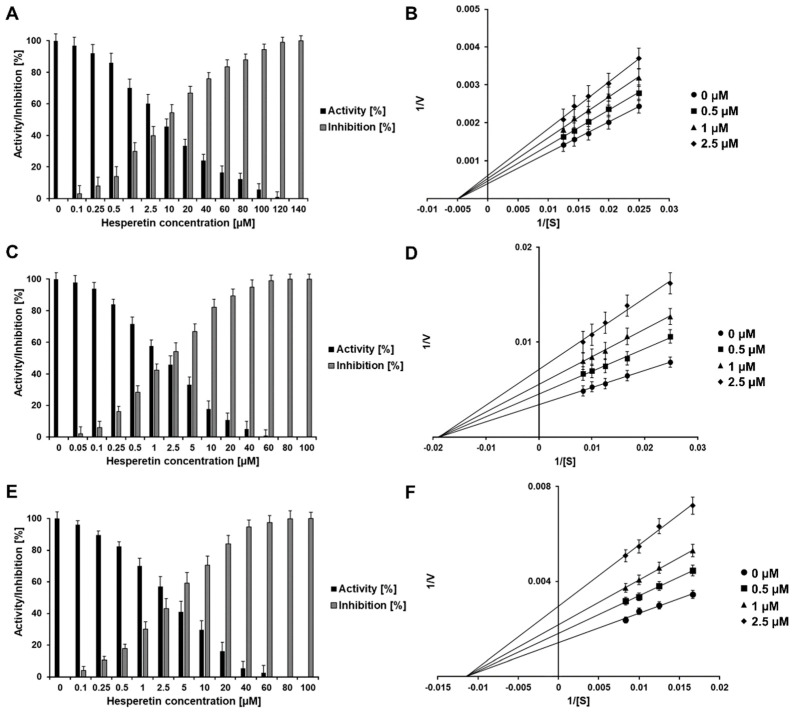
Normalized activity, inhibition effect and inhibition mode of HST over DENV2, YFV, and WNV NS2B/NS3^pro^. Lineweaver-Burk plots were used to determine the inhibition modes. [S] is the substrate concentration; v is the initial reaction rate. Inhibition of DENV2, YFV, and WNV NS2B/NS3^pro^ by HST are shown in (**A**,**C**,**E**), respectively. Lineweaver-Burk plots for HST inhibition of DENV2, YFV, and WNV NS2B/NS3^pro^ are shown in (**B**,**D**,**F**), respectively. Data shown are the mean ± SD from 3 independent measurements (*n* = 3).

**Figure 4 plants-10-02183-f004:**
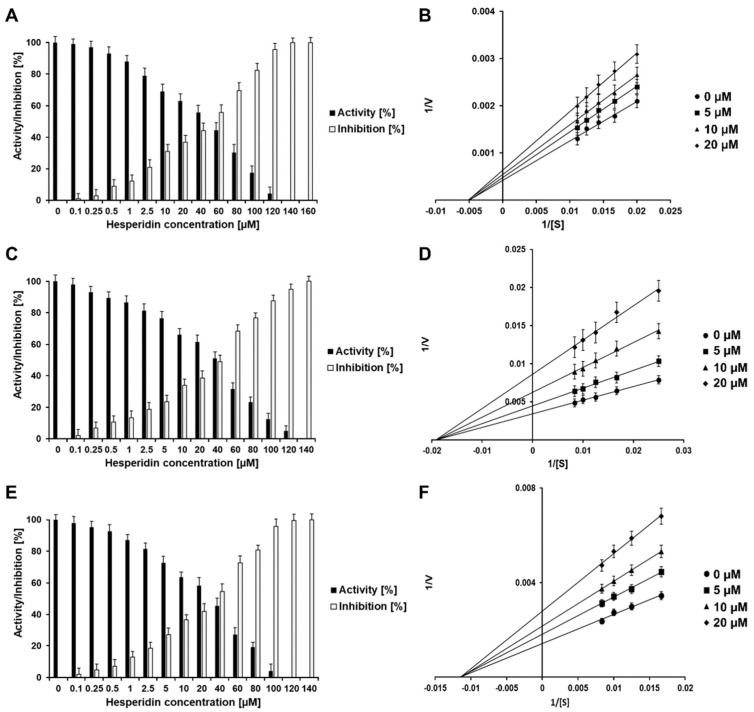
Normalized activity, inhibition effect and inhibition mode of HSD over DENV2, YFV, and WNV NS2B/NS3^pro^. Lineweaver-Burk plots were used to determine the inhibition mode of HSD. [S] is the substrate concentration; v is the initial reaction rate. Inhibition of DENV2, YFV, and WNV NS2B/NS3^pro^ by HSD are shown in (**A**,**C**,**E**), respectively. Lineweaver-Burk plot for HSD inhibition of DENV2 NS2B/NS3^pro^. Inhibition of YFV NS2B/NS3^pro^ by HSD is shown in (**B**,**D**,**F**). Data shown are the mean ± SD from 3 independent measurements (*n* = 3).

**Figure 5 plants-10-02183-f005:**
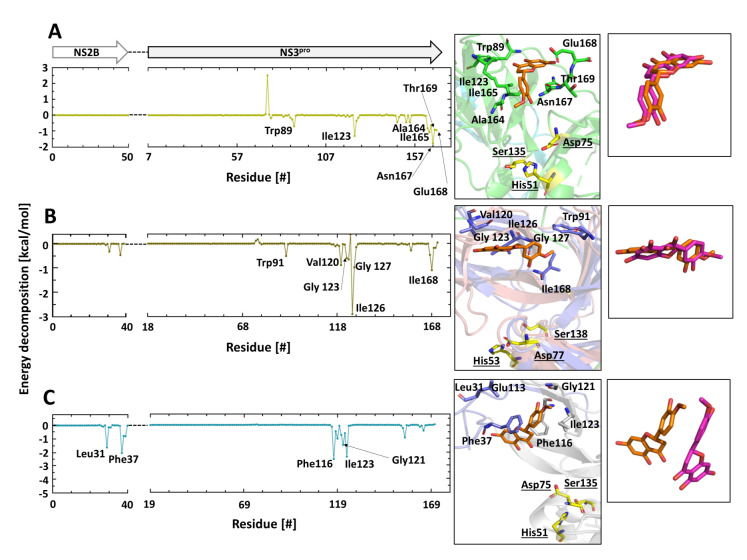
Amino acids contributing in the DENV2, YFV, and WNV NS2B/NS3^pro^–HST interaction. Decomposition energy of the amino acids that participated in the interaction with HST is based on the MD simulations. Coordination of HST in the protease binding region and overlay of the HST replicas 1 and 2: (**A**) DENV2 NS2B/NS3^pro^–HST interaction; (**B**) YFV NS2B/NS3^pro^–HST interaction; (**C**) WNV NS2B/NS3^pro^–HST interaction. Amino acid residues with an energy decomposition < −1 kcal/mol are considered.

**Figure 6 plants-10-02183-f006:**
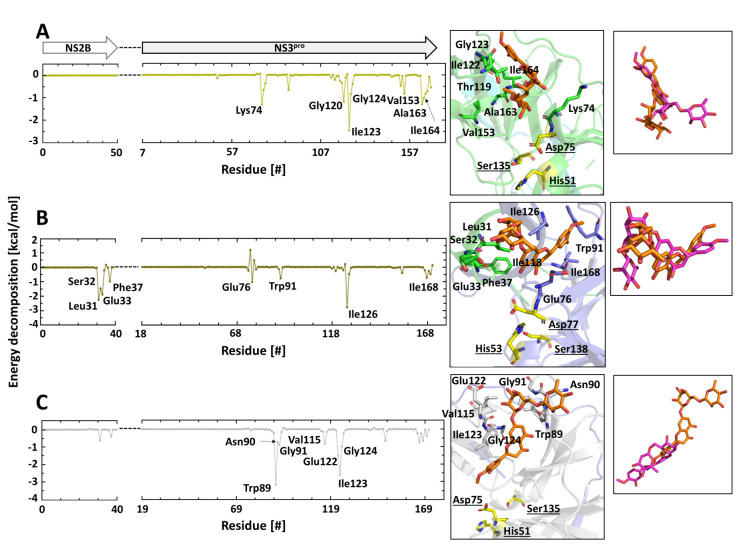
Amino acids contributing to the DENV2, YFV, and WNV NS2B/NS3^pro^–HSD interaction. The decomposition energies of amino acids that participated in the interaction with HSD are based on the MD simulations. Coordination of HSD in the protease binding region and overlay of the HSD replicas 1 and 2; (**A**) DENV2 NS2B/NS3^pro^–HSD interaction; (**B**) YFV NS2B/NS3^pro^–HSD interaction; (**C**) WNV NS2B/NS3^pro^–HSD interaction. Amino acid residues with an energy decomposition < −1 kcal/mol are considered.

**Table 1 plants-10-02183-t001:** Inhibitor key numbers and inhibition modes of HST and HSD for DENV2, YFV, WNV, and ZIKV NS2B/NS3^pro^.

Protease	Molecule	IC_50_ ^a^	Inhibition Mode	*p*IC50 ^b^	LE ^c^	N ^d^	Reference
DENV2 NS2B/NS3^pro^	HST	4.7 ± 0.8	noncompetitive	5.33	0.34	22	
	HSD	55.7 ± 2.5	noncompetitive	4.25	0.14	43	
YFV NS2B/NS3^pro^	HST	2.0 ± 0.5	noncompetitive	5.69	0.36	22	
	HSD	67.7 ± 7.5	noncompetitive	4.17	0.13	43	
WNV NS2B/NS3^pro^	HST	4.3 ± 1.6	noncompetitive	5.37	0.34	22	
	HSD	50.7 ± 8.3	noncompetitive	4.29	0.14	43	
ZIKV NS2B/NS3^pro^	HST	12.6 ± 1.3	noncompetitive	4.90	0.31	22	[11]
	HSD ^e^	-	-	-	-	-	[11]

^a^ IC_50_ value ± STD in µM. ^b^ Logarithm of IC_50_ value (*p*IC50). ^c^ Ligand efficiency (LE): LE > 0.3 suggests that the molecule is a potent lead compound. ^d^ Number of non-hydrogen atoms (N). ^e^ HSD was not further investigated in Eberle et al., 2021 [11].

**Table 2 plants-10-02183-t002:** K_D_ values of HST and HSD with DENV2, YFV, WNV, and ZIKV NS2B/NS3^pro^, as determined by TFS.

Protease	Molecule	K_D_ ^a^	References
DENV2 NS2B/NS3^pro^	HST	18.7 ± 2.7	
	HSD	20.9 ± 2.4	
YFV NS2B/NS3^pro^	HST	12.3 ± 2.2	
	HSD	29.5 ± 2.8	
WNV NS2B/NS3^pro^	HST	13.5 ± 2.0	
	HSD	19.5 ± 2.8	
ZIKV NS2B/NS3^pro^	HST	17.8 ± 2.9	[11]
	HSD ^b^	-	[11]

^a^ K_D_ value ± STD in µM. ^b^ HSD was not further investigated in Eberle et al., 2021.

**Table 3 plants-10-02183-t003:** Summary of the MD simulation replicas 1 and 2 of DENV2, YFVm and WNV NS2B/NS3^pro^ in complex (with HST and HSD molecules).

Protein + Ligand	Replicate	Binding Energy [kcal/mol]	Representative Frames Cluster [%]
DENV2 NS2B/NS3^pro^–HST	**1**	**−23.1 ± 3.6**	**68.9**
	2	−24.1 ± 3.7	62.0
YFV NS2B/NS3^pro^–HST	1	−20.0 ± 3.4	54.9
	**2**	**−19.4 ± 2.7**	**64.8**
WNV NS2B/NS3^pro^–HST	1	−20.2 ± 3.4	72.7
	**2**	**−28.3 ± 3.0**	**90.3**
DENV2 NS2B/NS3^pro^–HSD	**1**	**−37.4 ± 7.2**	**95.3**
	2	−38.6 ± 8.0	75.0
YFV NS2B/NS3^pro^–HSD	1	−35.4 ± 5.2	36.9
	**2**	**−37.6 ± 6.2**	**75.7**
WNV NS2B/NS3^pro^–HSD	**1**	**−40.7 ± 4.7**	**63.1**
	2	−37.8 ± 7.0	54.2

Chosen representative protease–ligand structure in bold.

## Data Availability

All data are reported in the text and Appendix A.

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
