# Peer review of "Promising Natural Compounds against Flavivirus Proteases: Citrus Flavonoids Hesperetin and Hesperidin"

_plants, 2021, doi:10.3390/plants10102183_

Round 1
Reviewer 1 Report
Flavonoids are important antiviral compounds. In this manuscript, the role of Hesperidin and Hesperetin in suppressing several flavivirus-coded proteases was investigated in vitro. The paper is well-organized and easy to follow. I just have few minor suggestions for the authors, as below.
1, the “HST and HSD” in the title should use the full name.
2, In the introduction, more information is needed to clarify how the project was initiated. In brief, how did the authors find that the HST and HSD inhibit the activity of the virus proteases?
3, Figure 1 is redundant, the structure of the two compounds was showed up again in Figure 2.
4, Though the interaction of HST / HSD and proteases were determined by TFS assay and MD simulation/prediction, I suggest the authors confirm this interaction using other experimental techniques, such as MST, if possible. It is well-adapted that at least two assays were needed to confirm a direct binding between two molecules, avoiding false positives.
Reviewer 2 Report
The article is really interesting, the mere idea acceptable, and I have only few remarks to consider by the authors in order to make the text clearer.
Comments and remarks:
Title: do not use abbreviations. It must be clear what it is about by reading the title itself.
Line 22: explain IC50 and KD when used for the first time, in abstract and in the main text.
Line 23-24: there is really no existing drugs nowadays based on those flavonoids?
Introduction: two first paragraphs are quite similar in the meaning (lines 28-39); make a shorter one paragraph.
Line 40: …; hesperidin (a small letter or in the entire text you will use capital H?); please decide.
Line64-69: this paragraph should be rewritten in order to provide a clear hypothesis (not conclusions) what the authors intended to do in that study. For instance: it was hypothesized that HST and HSD ….
Line 68: … and. (???)
Line 71-77: repetition; omit if possible.
Line 85: the phrase “there is no doubt…” sounds as no scientific language. Change.
Line 103: you used a Past tense “inhibited” and in the line 110: “inhibits” (Present tense); please unify and use a Past tense throughout the Result section.
Table 1 and table 2: you do not mention ZIKV in the title.
Line 212: 2.4[space] Molecular…
Statistics: I cannot see any statistics in figures 3, 4 and tables 1,2,3. Is it correct?
Conclusions: it must be shortened. For instance the lines 443-454 are not necessary – it is not introduction or abstract. Do not include references into conclusions.
References: please unify the journal cited names (abbreviations with dots or without) according to the Journal recommendations.
